# An Online Calibration Method Based on n-Tuple and Opportunistic Communication for Mine Mass Portable Gas Sensors

**DOI:** 10.3390/s21072451

**Published:** 2021-04-02

**Authors:** Gang Wang, Yang Zhao, Zeheng Ding, Xiaohu Zhao

**Affiliations:** 1IoT Perception Mine Research Center, China University of Mining and Technology, Xuzhou 221000, China; wanggang@cumt.edu.cn (G.W.); TS19060056A31@cumt.edu.cn (Y.Z.); TS20060122P31@cumt.edu.cn (Z.D.); 2The National Joint Engineering Laboratory of Internet Applied Technology of Mines, Xuzhou 221000, China; 3School of Information and Control Engineering, China University of Mining and Technology, Xuzhou 221000, China

**Keywords:** data fusion algorithm, gas monitoring data pairs, multi-dimensional data points, underground coal mine

## Abstract

Due to the increasing deployment of the Internet of Things (IoT) in the mining industry, portable gas monitoring devices have been widely used. Sensor calibration of large-scale portable gas monitoring devices is becoming an urgent problem to be solved. An online sensor calibration algorithm based on n-tuple and opportunistic communication is proposed based on the specific characteristics (i.e., ‘single-sensor, multi-position’ and ‘multi-sensor, single-position’) of each portable gas monitoring device employed. In this paper, data collected from portable and fixed sensors were defined as multi-dimensional data points and gas monitoring data pairs, respectively. The cluster-based self-adaptive weighted data fusion algorithm and multi-period single sensor reliability fusion algorithm were proposed and used for overall judging. The overall judgments were broadcast to each wireless access point by network, and the reliability of the calibration information transmission was enhanced by opportunistic communications. The simulation results revealed that efforts required for the calibration of portable sensors were reduced significantly, and their reliability was improved.

## 1. Introduction

The safety monitoring system for mining refers to a computer-facilitated system that monitors key environmental parameters (e.g., gas concentration, flow, CO concentration, smoke, and temperature) in addition to the working conditions of electromechanical equipment for various steps (e.g., production, transportation, elevation, and drainage). One of the key functions of a safety monitoring system is gas monitoring [1]. According to coal mine safety regulations, the mine chief, mining technology responsible person, blasting worker, excavating area leader, ventilation area leader, engineering and technical personnel, monitor, and flow electric fitter must wear portable gas detectors. Miners must wear helmets and lamps when they go down the mine. The portable gas detector is not necessary for every miner. Due to the increasing deployment of the Internet of Things (IoT) in the mining industry, portable gas monitoring devices (e.g., smart safety lamps with gas sensors) which could cover large areas and provide continuous monitoring have been widely utilized, and data transmissions from portable monitoring points to the control center are enabled. In a large mine, there are about a thousand people who work underground every day. The miner’s lamp and cap are the necessities for workers to work underground, and the gas sensors are just in the cap. Therefore, a large number of mobile gas detection devices are used every day. Gas monitoring is evolving from fixed point systems to fixed point plus portable device systems. However, this evolution is resulting in several issues. First, the correlation of data from the fixed sensors to those of portable ones should be clarified. Second, the calibration of a large quantity of portable sensors must be further investigated to ensure the reliability of data obtained. Third, the confidence level evaluation of the data from portable sensors should be fully understood due to their unique characteristics and ever-changing environments.

Most of these sensors show an electrochemical reaction when exposed to a specific gas. Ma H. et al. proposed a high-temperature silicon cantilever microheater with low power consumption to detect low concentrations of methane gas [2]. All these solid-state gas sensors are inexpensive, small, and suitable for mobile measurements. Hence, some researchers started to integrate them in mobile sensor nodes, such as smart safety lamps [3,4,5]. What’s more, wireless gas sensor networks (WGSN) for combustible or explosive gas monitoring are developed [6].

To assure high precision, the gas sensors are manually calibrated every few days. For example, in the Swiss National Air Pollution Monitoring Network (NABEL), the sensors are manually calibrated (adjusted in the parts-per-thousand range) every 14 days to assure high precision. The safety monitoring system for mining and instrument utilization norms (AQ1029-2007) provides detailed regulations for the installation, utilization, and calibration of gas sensors in safety monitoring systems. However, these regulations are not applicable to portable gas sensing systems. For instance, the sensor location requirement is not applicable to portable gas sensing systems; calibrating thousands of portable sensors every 10 d would result in a drastic workload increase. Unlike data collected by conventional fixed sensors, the data collected by portable sensors exhibits varying times and locations. Hence, data analysis and judging approaches for systems consisting of multiple gas sensors are urgently needed for gas monitoring in modern mines with IoT architectures. How to reduce the high workload of data calibration of large-scale portable gas monitoring devices is becoming an urgent problem to be solved [7,8].

Aimed at a bias load error calibration of weighting sensors and interactions of adjacent supporting points in this process, Yu XF et al. proposed a multi-sensor consistency calibration approach based on a normalization algorithm [9]. Niu et al. proposed a local sensor level decision threshold to maximize the system level deflection coefficient for a wireless sensor network with randomly deployed sensors [10]. Hasenfratz D et al. proposed three calibration algorithms such as forward, backward, and instant calibration that exploit co-located sensor measurements to enhance sensor calibration and consequently the quality of the pollution measurements on the fly [11]. Huang BK et al. proposed a 1D target-based approach for the global calibration of line-structured multi-sensor optical vision measurement systems, which are currently limited by the requirements of 3D measuring instruments and sophisticated calibration procedures [12]. Xue SH et al. proposed a multi-sensor fusion-based temperature calibration system for PCR instruments [13]. In this system, the setting temperatures of PCR instruments were calibrated based on a fusion of data collected from various sensors. Aiming at the Fusion Center (FC) for how to perform a more-accurate global decision, D. Ciuonzo et al. introduced the Generalized Likelihood Ratio Test (GLRT) to develop a novel fusion rule corresponding to a Generalized Rao (G-Rao) test based on Davies’ framework to reduce the computational complexity [14,15]. A neural network is a highly nonlinear platform and can perform complex calculations. Rongshan Wei et al. used an MCU-based sensor calibration system which mainly employed a particle swarm optimization (PSO)-back propagation (BP) neural network [16]. Dušan B. Topalović et al. demonstrated that data from each individual sensor system can be corrected using that sensor system’s own data to achieve much-improved data quality [17]. Abdolrahim et al. thought that the model-based calibrated gas sensor array could be a cheap alternative to other tools and this was tested in online monitoring [18]. Despite these works, few studies on portable gas sensor calibration in the mining industry have been reported.

In this paper, we propose automatic calibration algorithms can be used online to improve the measurement accuracy of portable low-cost gas sensors and reduce the high workload of data calibration of large-scale portable gas devices. This paper proceeds as follows. In Section 2, the performance of several commonly used gas sensors is compared, and the basic structure of the smart miner’s lamp is introduced. In Section 3, network models for fixed and portable monitoring systems are proposed. In Section 4, algorithms for two unique scenarios corresponding to fixed and portable gas sensors are proposed, and their conversion is demonstrated. Additionally, Section 5 details the ability of opportunistic communications to enhance the delivered gas calibration to improve the credibility of overall judgments. In Section 6, simulation tests of the proposed algorithms are presented. Finally, Section 7 concludes the paper.

## 2. Gas Sensor and Smart Miner’s Lamp

The smart miner’s lamp is worn on the miner’s helmet and is carried by the miner. It not only has the function of ordinary miner’s lamp, but also integrates the functions of gas monitoring and alarm, environmental temperature monitoring, and precise positioning of personnel. It can notify each miner of safety information in real time. The intelligent miner’s lamp builds a comprehensive perception and early warning system that actively perceives the coal mine environment for underground miners. Figure 1 shows the external and internal structure of the designed smart miner’s lamp. The intelligent miner’s lamp adopts the explosion-proof integrated MJC42/2.8 methane sensor to collect the gas concentration in the environment, its methane detection accuracy is ±0.10%, and the positioning accuracy is <5 m.

Gomes et al. [19] summarized several gas sensing technologies available for IoT-enabled gas sensors, as shown in Table 1.

Thermal catalytic sensors are currently the mainstream methane sensors for mines due to their simplicity, practicality, and low price. The MJC4/2.8J element uses the principle of catalytic combustion to work. The black and white element can be used for the concentration detection of flammable gases such as natural gas and alkanes at industrial sites. The two arms of the bridge are formed by pairing the detection element and the compensation element. When encountering flammable gas, the resistance of the detection element increases and the output voltage of the bridge changes. Therefore, the output voltage value reflects the methane change value. It has the characteristics of high activity and low power consumption. The service life of the sensor is 2 years. As the stability of the methane carrier catalytic element is greater than or equal to 7 d, the coal mine safety regulations formulated by the National Administration of Coal Mine Safety require that methane sensors and portable methane detection alarm devices using carrier catalytic elements must be adjusted with air samples and corresponding methane standard gas samples every seven days [20]. The adjustment of the mine gas sensor is divided into two steps: (1) zero-point adjustment, that is, in fresh air observe whether the sensor reading is 0; (2) in the standard gas sample (for example, the gas concentration is 2%), make sure the reading of the sensor is consistent with the concentration of the standard gas sample. Both of these scenarios will be encountered in actual mine surveys.

## 3. Network Models

Combined with the application of specific smart miner’s lamps, the gas monitoring network consisting of fixed and portable sensors based on the IoT architecture is shown in Figure 2.

A wireless access point with basic pre-analysis and judging programs for gas sensors was attached to each fixed gas sensor via a wired connection with a shared power line. As the miners with smart safety lamps that was equipped with a portable sensor passed by a fixed sensor, the data from the portable sensor was transmitted to the wireless access point attached to the fixed sensor. Then, the data collected by the wireless access point was pre-analyzed and successively transmitted to the ground work station. Usually, according to the network settings, the smart miner’s lamp will automatically record the gas, temperature, location and other information every 10s after the miner enters the mine, and of course the sampling time at this time will also be recorded. When the miner passes by the gas fixed detection point, since the gas fixed detection point also serves as a sink node (or wireless gateway node) function, the fixed gas node will collect the measurement data of the mobile gas node at this time together with the gas measured by itself. The overall judgment was implemented on the ground work station. The results of the overall judgment are broadcast to all access points (AP_1_ to AP_N_) and then delivered to fixed sensors to which the access point was attached as well as to portable sensors near the access point. The confidence level of the data from the portable sensors was investigated via a correlation analysis of portable sensors based on the fixed sensors in the local area, which is demonstrated in Section 3. When a decreased confidence level was encountered, a calibration was immediately conducted for the portable sensor using the IoT transmission platform and the adjacent access points to reduce the workload of routine calibrations. At last, calibration information is transmitted to each potable sensor by its attached access point.

Figure 3a is a miner wearing a smart miner’s lamp, Figure 3b is a picture of a fixed access point, and Figure 3c is the gas data displayed on the smart miner’s lamp panel.

When the miners are transported to the underground by the elevator, there may be several miners passing through the same point of access at the same time, as shown in the track 1 of the Figure 2. Taking the miner *k* as an example; there are many other miners nearby at every moment when they walk in the mine, and they pass through the same access point, such as the AP1-AP5 in the Figure 2. In addition, some of the miners may have walked through an access point on their own many times, such as miner l and the track 2 in the Figure 1, and no other miners and their peers are nearby when they pass the node AP_k_-AP_N_.

So, there are two unique scenarios attributed to downhole walking that may be observed for both fixed and portable gas sensors.

**Definition** **1.**
*Multi-sensor, single-position.*


Several sensors meet at a certain position at the same time; this scenario is denoted as ‘multi-sensor, single-position’. As shown in Figure 2, with each cluster in track 1.

This kind of scene usually takes place near the well head when the miner just goes down to the bottom of the mine. There are seven people in the cage at a time, but due to the gathering of people, there will be multiple miners at the same time with a fixed access point near the mine entrance.

**Definition** **2.**
*Single-sensor, multi-position*


Several positions were swept by the same sensor at different times; this scenario is denoted as ‘single-sensor, multi-position’. As shown in Figure 2 in track 2.

This kind of scene occurs when the miner moves underground. Because of the distance between the working point and the walking route, the miner will pass through different fixed access points at different times.

## 4. Algorithm Design

Since two unique scenarios attributed to downhole walking may be encountered, two spatio-temporal data structures were observed for the data from the fixed and portable gas sensors, and different algorithms shall be employed for the data fusion and correction of the portable gas sensors.

### 4.1. Cluster-Based Self-Adaptive Weighted Data Fusion Algorithm

In the ‘multi-sensor, single-position’ case, several measured values were received by a wireless access point at the same time and location, as shown in Figure 4. A multi-dimensional data point based on the values received can be described by an n-tuple or calibration tuple:(1)s=(x,y,t,ap)
where *x* is the raw data measured by the portable sensor, *y* is the data measured by the fixed sensor, *t* denotes the measuring duration, and *ap* denotes the location label. Because the portable sensor and fixed sensor appear to be at similar locations at similar times, we can combine them into a calibration tuple.

In the ‘multi-sensor, single-position’ case, the n-tuple or calibration tuple can be described as:(2)$$s_{k}=\{(x_{1k},\ldots ,x_{mk},y_{k},t_{k},ap_{k})|k=1,2,\cdots ,n\}$$
where $\left(x_{1k},\ldots ,x_{mk}\right)$ denotes the *m* values measured by the portable sensors near the fixed sensor *k* (*m* ≤ *M*, *n* ≤ *N*), *M* is the total quantity of portable sensors involved, *N* is the total quantity of fixed gas sensors, *y_k_* is the value measured by the fixed sensor *k*, *t_k_* denotes the measuring duration, and $ap_{k}$ denotes the location label.

We assume that when a miner passes through a fixed gas monitoring point, the gas monitoring concentration will not change significantly due to the difference in the distance between the miner and the fixed gas monitoring point. Only when the miner is far away from the fixed gas monitoring point will the monitoring value of the mobile gas sensor be significantly different. In other words, no sudden changes were expected for gas sensing, three successive values from the same portable sensor were used as the reference data, and the arithmetic average of these values was used as the measured value in this time window. Data analysis was conducted using a cluster-based self-adaptive weighted data fusion algorithm. *N* clusters corresponding to *N* fixed sensors were observed, and the intra-cluster calculation is as follows.

**Reasoning Process:** Cluster-based self-adaptive weighted data fusion algorithm.

**Step 1** Obtain the estimated value produced by each portable sensor.
(3)$${\overset{‒}{x}}_{k}=\frac{1}{m}\overset{m}{\underset{i=1}{\sum }}x_{ik}$$

**Step 2** Calculate the standard deviation of each portable sensor.
(4)$$\sigma_{ik}^{2}={\left(x_{ik}-{\overset{‒}{x}}_{k}\right)}^{2}$$

**Step 3** Determine the weight *w*_ik_ that minimizes the overall standard deviation of this cluster $f(w_{ik},\ldots ,w_{mk})=\Sigma w_{ik}^{2}\sigma_{ik}^{2}$, and obtain the weighting factor.
(5)$$w_{ik}=\frac{1}{\sigma_{ik}^{2}\underset{j}{\sum }\frac{1}{\sigma_{jk}^{2}}},\;i=1,2,\cdots m;\;j=1,2,\cdots ,m$$

**Step 4** Calculate the target parameters after intra-cluster fusion:(6)$${\hat{x}}_{k}=p_{1}\underset{i}{\sum }w_{ik}x_{ik}+p_{2}y_{k}$$
where *p*_1_ = 0.4 and *p*_2_ = 0.6. The value of *p*_1_ and *p*_2_ can be determined empirically as long as *p*_1_ + *p*_2_ = 1.

**Step 5** Define the confidence level of the portable sensor.
(7)$$C_{ik}=\frac{\left|x_{ik}-{\hat{x}}_{k}\right|}{\left|{\hat{x}}_{k}\right|},\;i=1,2,\cdots m$$

**Step 6** Define the overall confidence level of the *i*th portable sensor.
(8)$$C_{i}=C_{ik}$$

*C*_min_ was defined as the minimum confidence level of portable sensors in this study. In cases where *C_i_* > *C*_min_, the *i*th portable sensor needs to be calibrated.
$x_{ik}-{\hat{x}}_{k}$
was calculated and used as the calibrated offset of the *i*th portable sensor. Directional calibration of the *i*th portable sensor was achieved by a looped network and access points; the calibration time was labeled, and the calibration runs were increased by 1.

### 4.2. Multi-Period Single Sensor Reliability Fusion Algorithm

As shown in Figure 5, data pairs were obtained as a portable sensor swept different locations at different times. These data pairs can be described by:(9)$$s_{ik}=\left\{(x_{ik},y_{ik},t_{k},ap_{k})|k=1,2,\cdots ,n\right\}\;i=1,2,\cdots ,m$$
where *x_ik_* and *y_i__k_* denote the values measured by the *i*th portable sensor and *k*th fixed sensor, respectively, when the portable sensor was close to the fixed sensor. Additionally, *t_k_* and $ap_{k}$ denote the measuring time and location label, respectively. The pre-analysis was achieved using a multi-period single sensor reliability fusion algorithm, which went as follows.

**Reasoning Process:** Multi-period single sensor reliability fusion algorithm.

**Step 1** Data pairs corresponding to *n* randomly selected fixed gas sensors were included in a set denoted as *X*, and estimated values corresponding to the fixed and portable gas sensors were calculated based on the data pairs in this set:(10)$${\hat{x}}_{ik}=p_{1}x_{ik}+p_{2}y_{ik}$$
where *p*_1_ = 0.4 and *p*_2_ = 0.6. The value of *p*_1_ and *p*_2_ can be determined empirically as long as *p*_1_ + *p*_2_ = 1.

**Step 2** Define the confidence level of the *i*th portable sensor when it was close to the *k*th fixed sensor.
(11)$$C_{ik}=\frac{x_{ik}-{\hat{x}}_{ik}}{{\hat{x}}_{ik}},\;i=1,2,\cdots m;\;k=1,2,\cdots n$$

**Step 3** Define the overall confidence level of the *i*th portable sensor.
(12)$$C_{i}=\frac{1}{P}\overset{P}{\underset{k=1}{\sum }}C_{ik}$$
where *P* denotes the total appearance of the *i*th portable sensor in a set *X*.

*C*_min_ was defined as the minimum confidence level of portable sensors in this study. In cases where *C_i_* > *C*_min_, the *i*th portable sensor needs to be calibrated. The arithmetic average of $x_{ik}-{\hat{x}}_{k}$ was calculated and used as the calibrated offset of the *i*th portable sensor corresponding to *C_i_* > *C*_min_. Directional calibration of the *i*th portable sensor was achieved by a looped network and access points, and the calibration time was labeled. However, this calibration run should be void if another calibration run was observed in the same time window.

In this way, the confidence level and the calibration issues of the portable gas sensors were mitigated, and data analysis and judging in the presence of multiple gas sensors were achieved.

### 4.3. Relation between the Two Algorithms Proposed

The above two algorithms mainly focus on the route of the miners walking in the underground and the number of miners in the same place at the same time. The scenes of the miners in the underground are divided into two situations: multi-sensor, single-position and single-sensor, multi-position. The specific algorithm to choose depends on the gas data collected by the fixed gas detector and the mobile gas detector, and the miner’s underground movement trajectory can be analyzed. If there are multiple mobile gas detectors near the same fixed gas detector at the same time, Reasoning Process for Cluster-based self-adaptive weighted data fusion algorithm is adopted; if the miner experiences multiple different fixed gas detectors downstream of the well, Reasoning Process for Multi-period single sensor reliability fusion algorithm is adopted.

In essence, the multi-period single sensor reliability fusion algorithm and the cluster-based self-adaptive weighted data fusion algorithm can be conversed to each other.

**Theorem** **1.**
*The multi-period single sensor reliability fusion algorithm at n = 1 is essentially the cluster-based self-adaptive weighted data fusion algorithm at m = 1.*


**Proof.** Let *m* = 1, ${\overset{‒}{x}}_{k}=\frac{1}{m}\overset{m}{\underset{i=1}{\sum }}x_{ik}=x_{ik}$, $\sigma_{ik}^{2}={\left(x_{ik}-{\overset{‒}{x}}_{k}\right)}^{2}=0$ and $w_{ik}=1$, according to Equation (3). Thus, the target parameter after the intra-cluster fusion can be described as:$${\hat{x}}_{k}=p_{1}\underset{i}{\sum }w_{ik}x_{ik}+p_{2}y_{k}=p_{1}x_{ik}+p_{2}y_{k}$$
where *p*_1_ + *p*_2_ = 1. This result is consistent with Equation (10).Let *n* = 1, then *P* = 1, the overall confidence level of the *i*th portable sensor is
$$C_{i}=\frac{1}{P}\sum_{k=1}^{P}C_{ik}=C_{ik}$$This result is consistent with Equation (8). □

Therefore, it can be concluded that the multi-period single sensor reliability fusion algorithm is equivalent to the cluster-based self-adaptive weighted data fusion algorithm in two limiting cases, which shows the two algorithms are dialectical and unified.

### 4.4. A Joint Algorithm Based on the Joint Scene Proposed

The chosen of the proposed data fusion algorithm depends on the application scene whether it is “multi-point, single-position” or “single-point, multi-position”. In practical applications, there will be a situation to meet the two scenarios at the same time which is noted as “multi-sensor, multi-position”, for example “track 1” in Figure 2, and a data fusion algorithm based on the combination of the two scenarios is proposed.

**Reasoning Process:** A joint algorithm based on “multi-sensor, multi-position” case.

**Step 1** Obtain the estimated value produced by each portable sensor.
(13)$${\overset{‒}{x}}_{k}=\frac{1}{m}\overset{m}{\underset{i=1}{\sum }}x_{ik}$$

**Step 2** Calculate the standard deviation of each portable sensor.
(14)$$\sigma_{ik}^{2}={\left(x_{ik}-{\overset{‒}{x}}_{k}\right)}^{2}$$

**Step 3** Determine the weight *w_ik_* that minimizes the overall standard deviation of this cluster $f\left(w_{ik},\cdots ,w_{mk}\right)=\sum w_{ik}^{2}\sigma_{ik}^{2}$, and obtain the weighting factor.
(15)$$w_{ik}=\frac{1}{\sigma_{ik}^{2}\underset{j}{\sum }\frac{1}{\sigma_{jk}^{2}}},\;i=1,2,\cdots m;\;j=1,2,\cdots ,m$$

**Step 4** Calculate the target parameters after intra-cluster fusion:(16)$${\hat{x}}_{k}=p_{1}\underset{i}{\sum }w_{ik}x_{ik}+p_{2}y_{k}$$
where *p*_1_ = 0.4 and *p*_2_ = 0.6. The value of *p*_1_ and *p*_2_ can be determined empirically as long as *p*_1_ + *p*_2_ = 1.

**Step 5** Define the confidence level of the portable sensor.
(17)$$C_{ik}=\frac{\left|x_{ik}-{\hat{x}}_{k}\right|}{\left|{\hat{x}}_{k}\right|},\;i=1,2,\cdots m$$

**Step 6** Define the overall confidence level of the *i*th portable sensor.
(18)$$C_{i}=\frac{1}{P}\overset{P}{\underset{k=1}{\sum }}C_{ik}$$
where *P* denotes the total appearance of the *i*th portable sensor corresponding to n fixed sensors.

If *C_i_* > *C*_min_, the arithmetic average of $x_{ik}-\;{\hat{x}}_{k}$ was calculated and used as the calibrated offset of the ith portable sensor corresponding to *C_ik_* > *C*_min_.

The specific implementation of the process is shown in Algorithm 1.
**Algorithm 1.** A joint algorithm based on “multi-sensor, multi-position” case.**Input**: (*x*_1*k*_, …, *x*_*mk*_), *y_k_*, *t_k_*, *ap*_*k*_, *k* = 1,2,…,*n*, *C*_min_
**Output**: *C_i_*, Calibration parameter $\Delta x_{i}$
1: Compute the estimated value ${\bar{x}}_{k}$ through Equation (13)

2: Compute $\sigma_{ik}^{2}$ and weight *w_ik_* through Equations (14) and (15)

3: Compute the intra-cluster fusion value ${\hat{x}}_{k}$ through Equation (16)

4: Compute the confidence level *C_ik_* of the portable sensor through Equation (17)

5: Compute the overall confidence level *C_i_* through Equation (18)

6: **If**
*C_i_* > *C*_min_
$\Delta x_{i}=x_{ik}-{\hat{x}}_{k}$**End If**

Obviously, the Reasoning Process for Cluster-based self-adaptive weighted data fusion algorithm is a special case of the Algorithm 1 at *n* = 1, and the Reasoning Process for Multi-period single sensor reliability fusion algorithm is a special case of the Algorithm 1 at *m* = 1.

## 5. Broadcast of Correction Information and Opportunistic Communications

Calibrated offsets of the gas sensors of the portable sensors were broadcast via a looped network and access points. Upon receiving the calibration information, a confirmation message was sent and the output values were corrected accordingly. Further, the calibration time was labeled and the calibration runs were increased by 1.

Due to the random downhole routes of workers, it is possible for portable sensors to be disconnected from the network at certain moments. In this case, calibration information cannot be received by the sensor. To avoid this, opportunistic communications can be deployed. Figure 6 illustrates the opportunistic network communication [21]. For instance, at *t* = *t*_1_, calibration information needs to be transmitted from access point S to portable gas sensor D, which was not connected to the network. S and D were located in different connection domains, and there was no communication path between them. Briefly, a message containing the calibration information was sent to Node 1 and 2 and stored. At *t* = *t*_2_, Node D approached the communication range of Node 2, and then, the message was sent to portable gas sensor D. In this way, data transmission from access point S to portable gas sensor D was achieved.

Opportunistic networks originated from delay-tolerant networks that were commonly used in the early days, and now they are widely used for the connections and interactions of networks characterized by intermittent connections, significant delays, and high error rates [22,23]. Correction information transmission procedures via opportunistic communications were as follows:

The correction information received by a portable gas sensor was stored, carried, and re-transmitted at an appropriate moment, which was realized by node movements.

Retransmissions by nodes were facilitated by the active movements of these nodes. For intermittent connection-based communications, there is a significantly higher possibility of information transmission between adjacent portable sensors than between portable sensors that are remotely located. Thus, the portable sensors near the access point corresponding to the last appearance of a missing node would actively search for the missing node to provide connections to this node.

Upon receiving the correction information, the missing node sent a confirmation message to the control unit, thereby completing this re-transmission cycle.

In this way, the reliability of the calibration information transmission was enhanced by opportunistic communications.

## 6. Experiments and Test

In mine IoT applications, in order to balance personnel positioning accuracy and cost, the distance between two wireless access points is generally greater than 100 m. Of course, this depends on the requirements of underground personnel positioning accuracy. In this experiment, the distance between the two wireless access points is set to 150 m. Since the gas in the mine tunnel changes slowly, the gas measurement interval is generally selected as 10 s. According to the moving speed of the miner, the point closest to the wireless access point is selected to record the data of the mobile gas sensor and the fixed gas sensor.

### 6.1. Algorithm and Calibration Results

Taking 12 miners with smart safety lamps as a test, the 12 portable sensors approach each fixed sensor simultaneously. That is, the miners move along the track 1 route. Gas trends (the change trend of the gas concentration along the access point of the roadway, in %) obtained from twelve portable sensors and the fixed sensor are shown in Figure 7. The number of measured points is 188. Since the fixed gas monitors are all hung on the wall of the tunnel, when a certain place of gas protrudes slightly, due to the ventilation of the tunnel, the concentration will gradually decrease as it is farther away from the protruding point, thus showing a downward trend. Some sensors have obvious peaks at some measuring points, indicating that a measurement error has occurred here. This may be one of the following situations [24]: (1) The sampling of the mobile gas sensor is not synchronized with the fixed gas sensor, and does not reflect the gas data at the same time; (2) When sampling, the mobile gas sensor is too far away from the fixed gas sensor. Does not reflect the gas situation at the same location; (3) Due to the harsh mine environment and serious interference, the weak signal output by the sensor is easily interfered, resulting in abnormal data; (4) When a catastrophe occurs in a certain area of the mine, due to the maximum value of gas accumulation, gas anomalous data will be generated at this time; (5) Measurement errors caused by gas power-down or power-on. These values will be removed in the subsequent analysis.

After transmitting the data pair measured by the fixed gas sensor and the mobile gas sensor to the ground workstation, since the fixed gas sensor corresponds to the specific location of the roadway, the workstation first analyzes the underground behavior of the miner based on the received data pair and sorts out the underground action route of the miner. According to the number of miners at the same time, at the same location, or at the same time at different locations, select the corresponding calibration Reasoning Process for Cluster-based self-adaptive weighted data fusion algorithm, Reasoning Process for Multi-period single sensor reliability fusion algorithm or Algorithm 1. The diagnosis algorithm analyzes the measurement error of the gas sensor based on the gas measurement data and the measurement data of other measurement nodes at the same time and at the same location or the measurement data of the same gas measurement node at different locations to calibrate the gas sensor.

The algorithm calibrates the system error of the mobile gas monitor through the network based on the number of data pairs formed by the fixed gas detector and the mobile gas detector in a calibration cycle. Therefore, the gas data measured outside the access point will not be used as calibration reference data. In principle, only when the mobile gas detector is closest to the fixed gas monitor, the measurement data obtained by the two at the same time and at the same place will be used as the calibration reference data.

Figure 8, Figure 9, Figure 10 and Figure 11 show the gas trends of the measured values of the mobile gas detector, fixed gas measurement value, the mean value, the fusion value within the cluster and the calculated confidence of each measured value using Reasoning Process for Cluster-based self-adaptive weighted data fusion algorithm. Because there are 188 fixed stations, 188 clusters are formed and there are 188 different confidence levels. If *C*_min_ = 0.2 is defined as the minimum confidence level of portable sensors in this study, it can be seen from the figure, the measured value from portable sensor 3, 4, and 8 are much lower than the fixed gas sensor readings and their confidence levels are more than 0.2, so these sensors need to be calibrated.

Figure 12 shows the normalized confidence levels of 12 channel signals based on Reasoning Process for Multi-period single sensor reliability fusion algorithm. It can be seen from the figure that the measured values from portable sensor 3 and 8 are higher, so these sensors need to be calibrated. In addition, portable sensor 2, 4, 12 are also higher; this conclusion is different from the above drawn by Reasoning Process for Cluster-based self-adaptive weighted data fusion algorithm. The conclusion is due to some errors in data, which have an impact on the analysis results.

Figure 13 shows the normalized confidence levels of 12 channel signals based on Algorithm 1. It can be seen from the figure that the measured values from portable sensor 3, 4, and 8 are higher, so these sensors need to be calibrated. The conclusion is the same as the above drawn by Reasoning Process for Cluster-based self-adaptive weighted data fusion algorithm which shows that Algorithm 1 has the ability to suppress abnormal data points.

To calibrate the portable sensors by internet, a clustering of data from all of the sensors was achieved using the short-term gas flow fuzzy C average value clustering algorithm at first to determine which range of gas data was stable, and shared stabilized gas curves were used for further analysis [25]. Based on the clustering results, gas sequence samples 72-121 of 1-188 are selected for further analysis, resulting in *N* = 50 and *m* = 12 in this study. Data corrections of these sensors using the cluster-based data fusion algorithm are conducted, and the results are detailed in Figure 14. More specifically, Figure 12a shows the confidential level obtained. With *C*_min_ = 0.4, Sensors 3, 4, and 8 should be calibrated. Figure 14b shows the calibrated offsets of the gas sensors. It can be seen that the offsets of sensors 3, 4, and 8 were −0.068, −0.058, and −0.144, respectively. The calibrations were consistent with actual gas curves.

Figure 15 is the results of the calibrated 12 portable sensors using the multi-period multi-sensor reliability fusion algorithm proposed in algorithm. From the figure, the data measured by the portable sensors are closer to achieving the goal of adjusting the sensor through the network.

### 6.2. Algorithm Confidence and Systematic Error

Figure 16 and Figure 17 show the trustworthiness and systematic error of the algorithm. For each type of experiment, we consider a Monte Carlo method to run 100 experiments under the same parameters. For Reasoning Process for Cluster-based self-adaptive weighted data fusion algorithm, this involves taking a fixed gas monitoring point as an example by changing the number of mobile gas nodes in the same location at the same time. Figure 16a shows the distribution of confidence level. Figure 16b shows that when the number of nodes changes, the system error calibrated by Reasoning Process for Cluster-based self-adaptive weighted data fusion algorithm decreases with the increase of gas nodes.

For Reasoning Process for Multi-period single sensor reliability fusion algorithm, we fixed the mobile gas sensor and changed the number of fixed gas access points. The experiment results were the same as Reasoning Process for Cluster-based self-adaptive weighted data fusion algorithm. With the increase of the number of fixed gas access points, the systematic error was also reduced. According to Figure 16 and Figure 17, in order to achieve a relative error of less than 4%, the number of mobile gas sensors or fixed access points should be more than 7.

## 7. Conclusions

Based on the characteristics (i.e., ‘single-sensor, multi-position’ and ‘multi-sensor, single-position’) of portable gas monitoring devices, the data collected from portable and fixed sensors were defined as multi-dimensional data points and gas monitoring data pairs, respectively. The cluster-based self-adaptive weighted data fusion algorithm and multi-period single sensor reliability fusion algorithm were proposed and used for gas measurement calibrations which reduced the calibration workload significantly. The overall judgments were then broadcast to each wireless access point by network, and the reliability of the calibration information transmission was enhanced by opportunistic communications. This algorithm is especially suitable for calibration and maintenance of equipment in online operation. Other calibration errors of calibration can be further analyzed in the later stage.

## Figures and Tables

**Figure 1 sensors-21-02451-f001:**
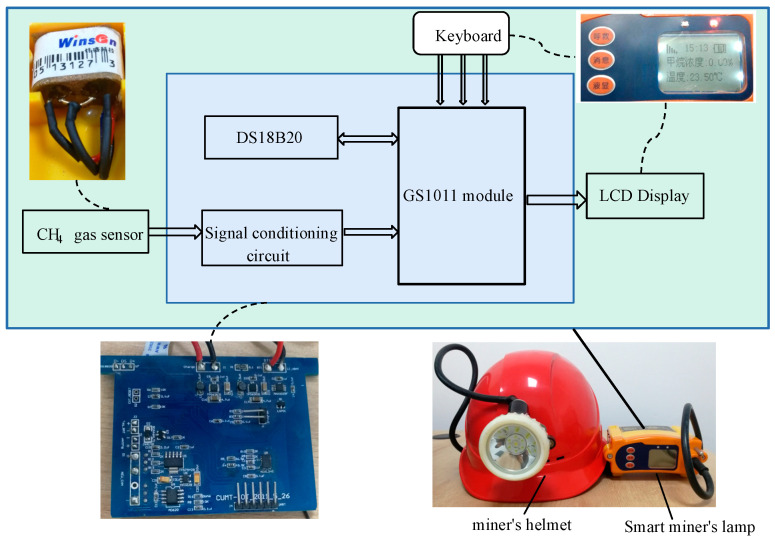
Smart miner’s lamp.

**Figure 2 sensors-21-02451-f002:**
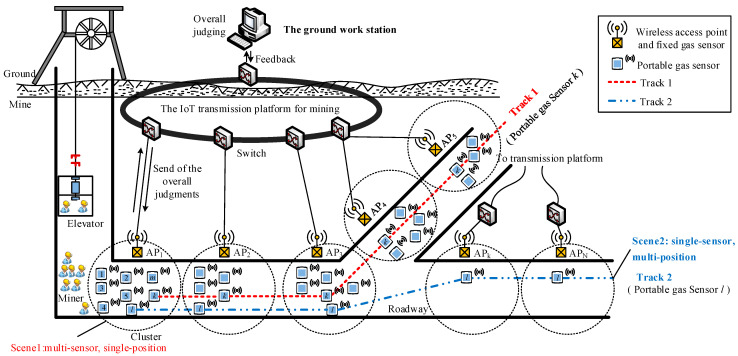
Network model of monitoring system.

**Figure 3 sensors-21-02451-f003:**
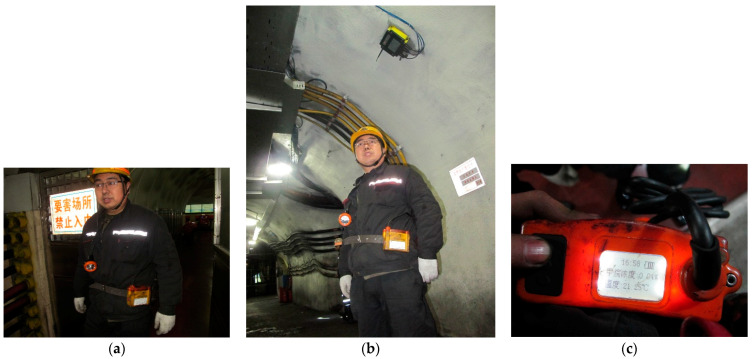
Network model of monitoring system. (**a**) Miner and smart miner’s lamp, (**b**) fixed gas sensor and wireless access point, (**c**) gas monitoring data.

**Figure 4 sensors-21-02451-f004:**
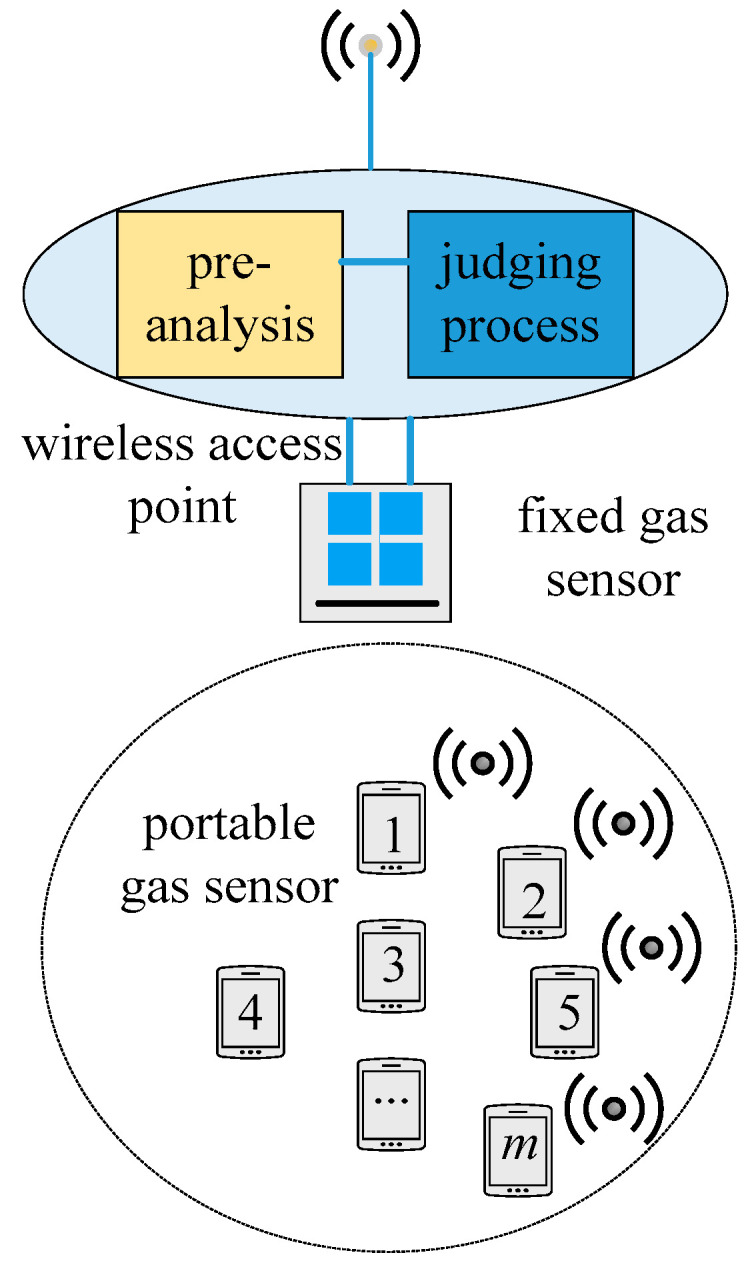
Scene of “multi-point, single-position”.

**Figure 5 sensors-21-02451-f005:**
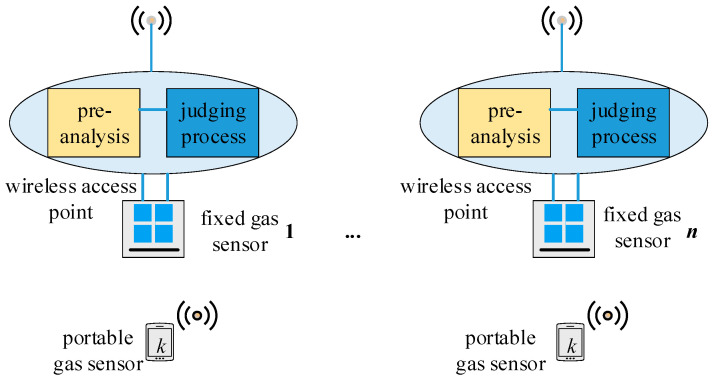
Scene of “single-point, multi-position”.

**Figure 6 sensors-21-02451-f006:**
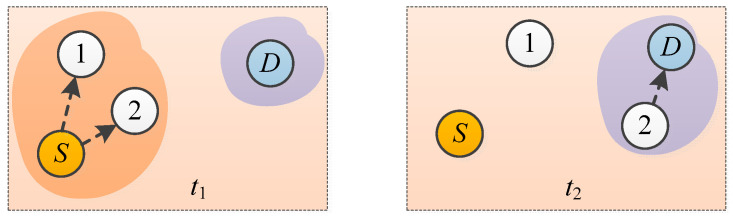
Communication of opportunistic networks.

**Figure 7 sensors-21-02451-f007:**
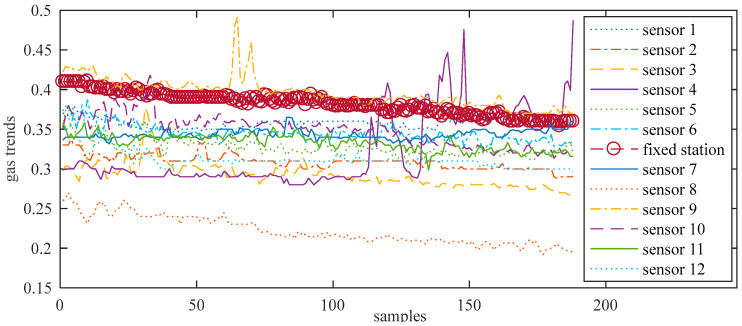
12 channel test gas signal and fixed gas signal.

**Figure 8 sensors-21-02451-f008:**
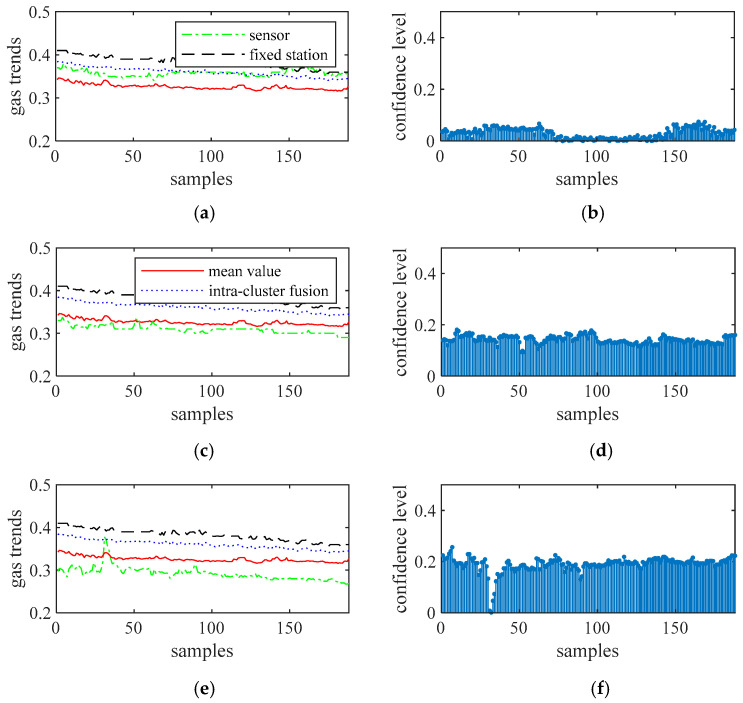
The measured values of sensors 1 to 3 and their confidence level. (**a**) gas sensor 1, (**b**) confidence level of gas sensor 1, (**c**) gas sensor 2, (**d**) confidence level of gas sensor 2, (**e**) gas sensor 3, (**f**) confidence level of gas sensor 3.

**Figure 9 sensors-21-02451-f009:**
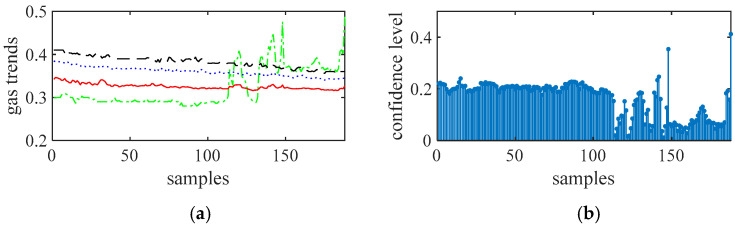
The measured values of sensors 4 to 6 and their confidence level. (**a**) gas sensor 4, (**b**) confidence level of gas sensor 4, (**c**) gas sensor 5, (**d**) confidence level of gas sensor 5, (**e**) gas sensor 6, (**f**) confidence level of gas sensor 6.

**Figure 10 sensors-21-02451-f010:**
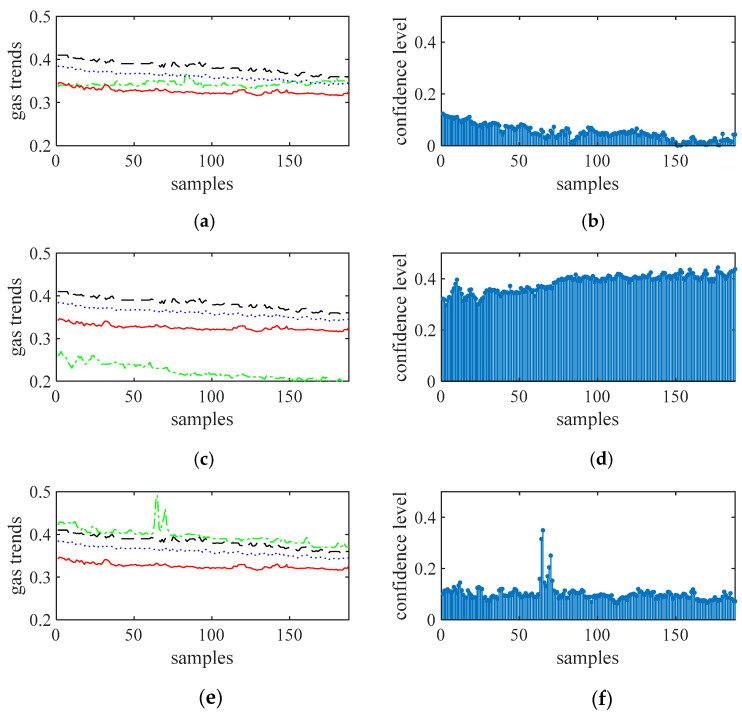
The measured values of sensors 7 to 9 and their confidence level. (**a**) gas sensor 7, (**b**) confidence level of gas sensor 7, (**c**) gas sensor 8, (**d**) confidence level of gas sensor 8, (**e**) gas sensor 9, (**f**) confidence level of gas sensor 9.

**Figure 11 sensors-21-02451-f011:**
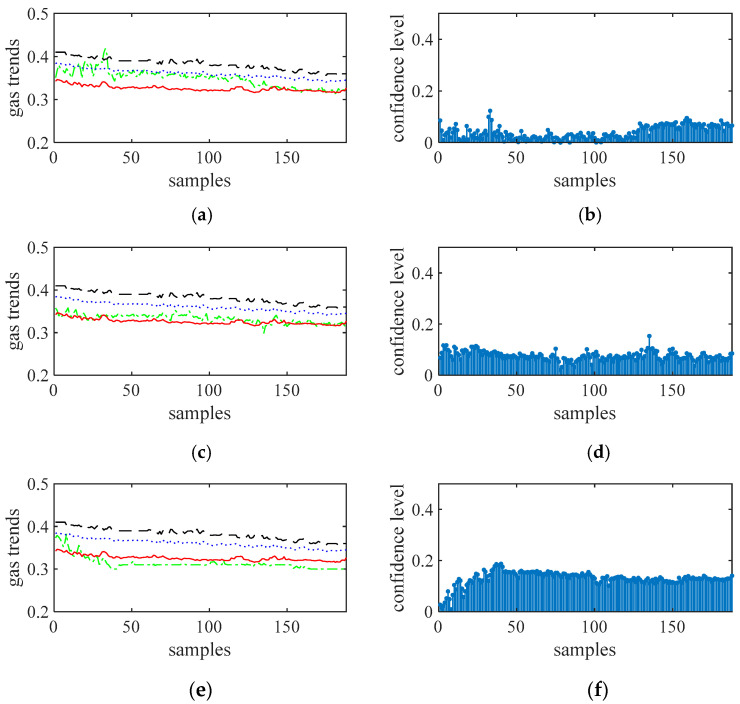
The measured values of sensors 10 to 12 and their confidence level. (**a**) gas sensor 10, (**b**) confidence level of gas sensor 10, (**c**) gas sensor 11, (**d**) confidence level of gas sensor 11, (**e**) gas sensor 12, (**f**) confidence level of gas sensor 12.

**Figure 12 sensors-21-02451-f012:**
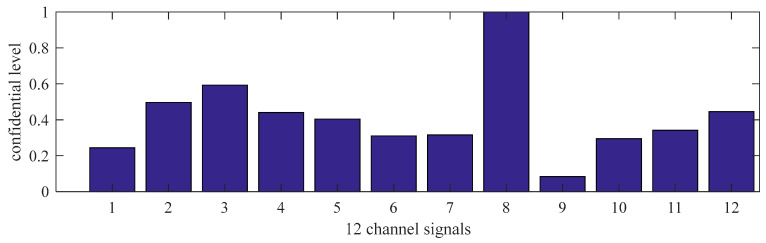
The confidence levels of 12 channel signals based on Reasoning Process for Multi-period single sensor reliability fusion algorithm.

**Figure 13 sensors-21-02451-f013:**
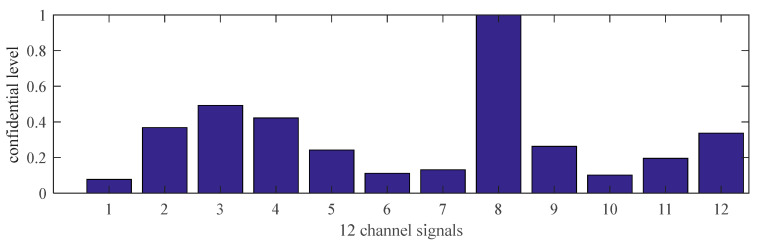
The confidence levels of 12 channel signals based on Algorithm 1.

**Figure 14 sensors-21-02451-f014:**
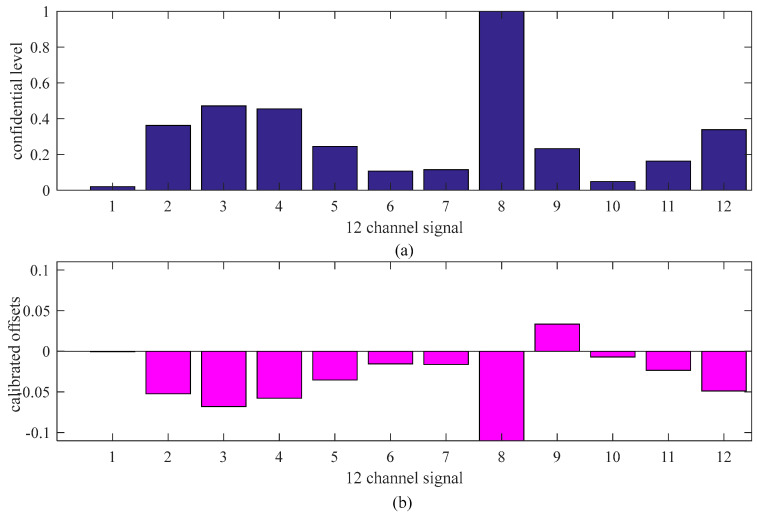
Gas data correction based on the adaptive weighting data fusion algorithm. (**a**) conficential level, (**b**) calibrated offsets.

**Figure 15 sensors-21-02451-f015:**
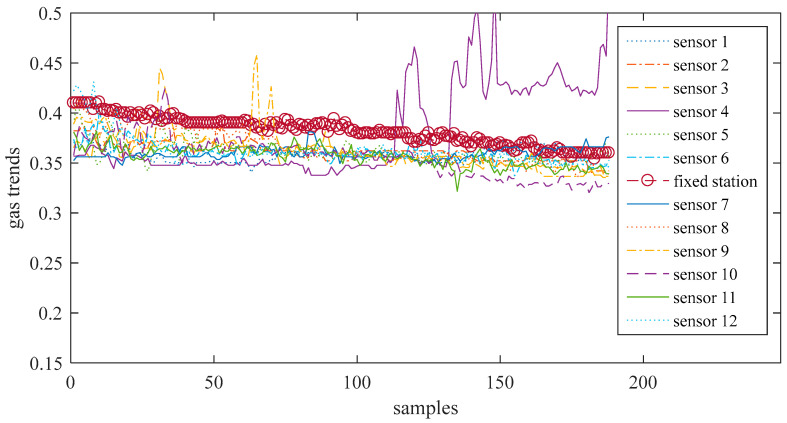
The calibrated 12 channel test gas signal and fixed gas signal.

**Figure 16 sensors-21-02451-f016:**
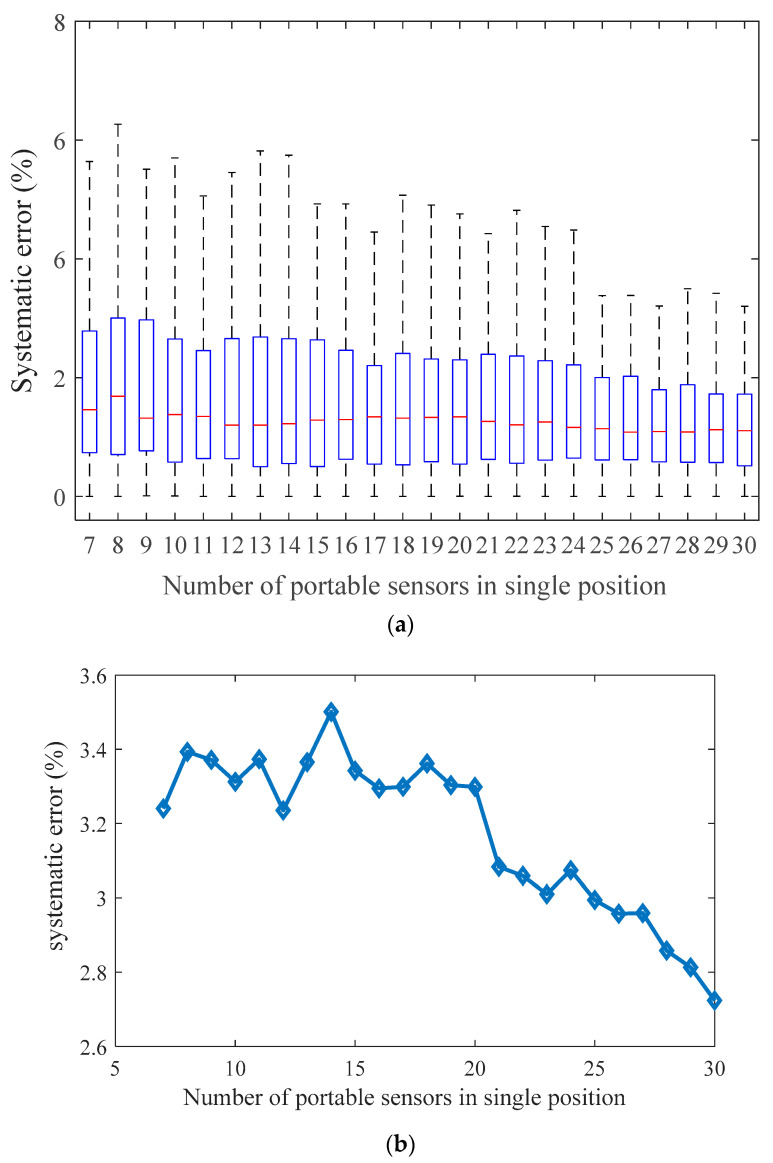
Trustworthiness and average systematic error of Reasoning Process for Cluster-based self-adaptive weighted data fusion algorithm. (**a**) Trustworthiness, (**b**) Average systematic error.

**Figure 17 sensors-21-02451-f017:**
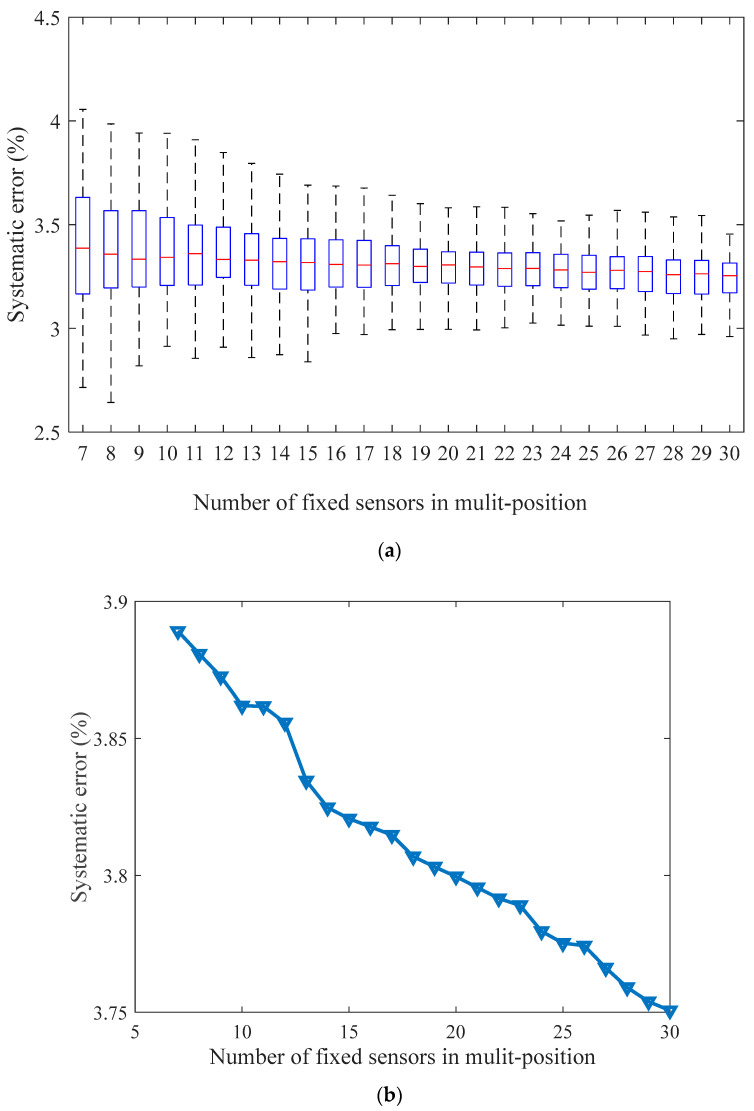
Trustworthiness and average systematic error of Reasoning Process for Multi-period single sensor reliability fusion algorithm. (**a**) Trustworthiness. (**b**) Average systematic error.

**Table 1 sensors-21-02451-t001:** Several gas sensing technologies available for IoT-enabled gas sensors.

Technology	Lifetime	Features
Electrochemical	2–5 years	Interference from other gases and environmental factors Low sensitivity and selectivity High response time
Metal Oxide Semiconductors (MOS)	2–5 years	Need of heaters Interference from other gases Low sensitivity and selectivity
Thermal Catalytic	<2 year	Need for oxygen to perform gas detection Short adjustment cycle Low price
Polymers	<6 months	Instability Long response and recovery times
Infrared	5–8 years	Long adjustment period Large sensor size High cost
Optic	5–10 years	Difficulty in miniaturization and high cost

## Data Availability

The datasets in this paper can be obtained from the following link: https://github.com/waitf10/n-Tuple-and-Opportunistic-Communication (accessed on 5 March 2021).

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
