# Peer review of "An Online Calibration Method Based on n-Tuple and Opportunistic Communication for Mine Mass Portable Gas Sensors"

_sensors, 2021, doi:10.3390/s21072451_

Round 1
Reviewer 1 Report
An Online Calibration Method Based on n-tuple And Opportunistic Communication For Mine Mass Portable Gas Sensors
As things become more connected in the mining industry, the use of mobile sensor arrays for safety is becoming attractive. The authors present an algorithm that aims to address the problem of calibrating mobile sensors while in operation.
The authors argue that a main issue with mobile sensors, is the increased workload resulting from the regular calibration of the sensors. Here it should be clarified why there would be a larger number of mobile sensor modules than traditional stationary systems as the mobile ones are more precisely placed. Here it would be important to discuss more thoroughly the different types of sensors that can be used and their long term stability. Is there literature that examines the ‘’real’ stability of the used sensors in application relevant conditions? How often is a calibration expected to be necessary? Would a daily normalization of the baseline, for example, when the miner goes home from work and leaves the lamp in fresh outdoor air, be enough?
Overall the paper provides far too little information about the measurement setup, and used sensor type (variance between new sensors). This is crucial for the paper. How does the hosing of the fixed sensors compare to that of the mobile sensors? Here it would be incredibly helpful for the reader to have a visual aid of the actual measurement setup.
‘‘As no sudden changes were expected for gas sensing…‘‘ how often are the portable sensors measured? In the gas of a gas release/ leak, depending on what type of sensor is being used, there could be a very significant change in the signal as result of gas exposure, can this be taken into account in this procedure? This could be more relevant in the case of the Multi-period single sensor reliability fusion algorithm? How is the need for calibration differentiated from a gas event?
Figure 5 is a bit confusing. What exactly is plotted in the y-axis? How was the ‘‘gas trend‘‘ determined from the raw sensor response? What are the spikes in the gas trends visible for some of the sensors (notably sensor 4 and sensor 7)? The fixed position gas sensor is shows a consistently decreasing ‘‘gas trend.‘‘ What is the origin of this drift?
More information is needed about how the sensors are calibrated? In Figure 13, it appears as though the ‚‘‘baseline gas trend‘‘ has been normalized, but how do the ‚‘‘calibrated‘‘ sensors respond to a gas event? Often baseline drifts are indicative of overall sensor degradation (decrease in the sensor response), it is unclear how the suggested calibration method would address the decrease in sensor response over time. Also here it is vital that more information about the sensor type is provided.
The manuscript would benefit from an English correction and from a careful proofreading in order to correct typos (missing spaces, mart instead of smart on pg.3, incorrect spelling of Algrithm in figure captions).
Author Response
Dear review expert,
We should like to express our appreciation to you and the referees for suggesting how to improve our paper. All the modifications are marked in blue fonts. The following is the response to review. Please see the attachment.
Best Regards,
Gang Wang, Yang zhao ,Zeheng Ding and Xiaohu zhao

Reviewer 2 Report
This paper presents an on-line calibration algorithm for a network of gas sensors, in use in the mining industry.
The network consists of mobile as well as fixed sensors.
The algorithm is presented in detail but it is hard to follow the details.
The work is sophisticated and of great importance but could it be presented in a more easier format to the reader? For example, showing main results while the proof is presented in an Appendix?
Assuming the algorithm is innovative and correct, the reviewer is asking:
- What is the confidence level for each algorithm when compared with real collected data. Can the authors report the real error?
- How much is the calibration accuracy degraded during the period of being outside the access point, in particular for the situation of single sensor, multi-position scenario? For example the offset may be affected by the location below the earth.
- How long can a worker stay with one sensor beyond the access point?
- What is the minimal number of mobile sensors required to get adequate confidence level?
The reviewer is aware that the questions refer to system issues rather than the algorithm.
In summary, the manuscript presents high-level work but for the reader the material is very difficult to follow.
Could the paper be rewritten in a more friendly approach?
Author Response
Dear review expert,
We should like to express our appreciation to you and the referees for suggesting how to improve our paper. All the modifications are marked in blue fonts. Please see the attachment.
Best Regards,
Gang Wang, Yang zhao ,Zeheng Ding and Xiaohu zhao

Reviewer 3 Report
Authors have to consider the following points.
How to define the noises of confidence level? In Figs. 7(b) and 8(f), some noises are observed. How did authors treat the noises of confidence level?
Lines 345-346, authors state that the conclusion is due to some errors data, which have an impact on the analysis results. How to judge the data is error or not? In Fig. 10, how to tell which confidence levels are bigger? What is the limit? and authors have to state the reasons in detail.
In my point of view, this article is based on n-tuple and opportunistic communication; however, it is very unclear that how the system can be applied to the real site. For example, the standard gas must be used in the calibration system? And how to connect the signals between the calibration system and sensor? Authors must state the application approach for the online calibration method.
Author Response
Dear review expert,
We should like to express our appreciation to you and the referees for suggesting how to improve our paper. All the modifications are marked in blue fonts. The following is the response to review.
Best Regards,
Gang Wang, Yang zhao ,Zeheng Ding and Xiaohu zhao

Round 2
Reviewer 1 Report
Through the revisions the quality of the manuscript has significantly increased, there are still however a few clarifications needed.
In their answer to Q5 the authors state that, ‘‘Gas can have very obvious changes. In the follow-up tests, the signals tested by many sensors are not stable, which shows this problem. ‘‘What do the authors mean by this? Would a high variance in the response of the sensors to relevant gases not be detrimental to the entire setup? Are all the sensors replaced at the same time ensuring that they are of similar manufacturing batches to reduce sensor to sensor variation (every 2 years) or will the replacements be stacked in order to ensure that at least some of the sensors are still in ideal operating condition?
As under ideal working conditions the number of dangerous, ‚‘‘gas exposure events‘‘ should be minimal, it is unclear if the suggested calibration method is sufficient (the sensor‘s response to gas concentration relation is adjusted over time). Will the sensor respond in an expected manner to a sudden increase in e.g. methane. Catalytic conversion type sensors often suffer from a baseline drift and a decreased reactivity i.e. a slow agglomeration of the catalyst on the surface. The change in baseline however doesn’t necessarily correlate directly with the change in the sensor response, meaning that potentially the baseline can barely change but the sensor‘s response to an combustible gas could be significantly degraded. Would the calibration suggested by the authors ensure that the decrease in sensor response with time is accounted for and that an alarm is still activated at the right gas threshold concentration? Here it would be necessary for the reader to know how the concentration output of the sensor is attained. Is it based on a sensor signal (value during gas exposure referenced to a zero value) or is it simply assumed a certain voltage value indicates a certain gas concentration. Ideally the authors should include data in the manuscript from at least one, ‚‘‘gas exposure‘‘ event, or use an different measurement method to verify that overtime the concentration of methane determined by the sensors overall (in a certain area of the mine) is the actual methane concentration.
On page 3 the authors offer a very genera overview of a large number of papers. The added value of this section is limited. Here it would be better if the author concentrated on papers more relevant to topic, i.e. https://www.mdpi.com/2224-2708/8/4/57
There are still several typos in the manuscript, e.g. pg.4 line 161.
Author Response
Dear review expert,
Thank you for your affirmation of the revised manuscript. At the same time, we should like to thank you for your further comments on the paper, which will improve the quality of the paper again. All the modifications are marked in red fonts. The following is the response to review.
Best Regards,
Gang Wang, Yang Zhao ,Zeheng Ding and Xiaohu Zhao

Reviewer 3 Report
The manuscript is revised completely and it can be published in the current form.
Author Response
Dear review expert,
Thank you for your affirmation of the revised manuscript. All the modifications are marked in red fonts.
Best Regards,
Gang Wang, Yang Zhao ,Zeheng Ding and Xiaohu Zhao